# Effective Infection with Dengue Virus in Experimental Neonate and Adult Mice through the Intranasal Route

**DOI:** 10.3390/v14071394

**Published:** 2022-06-26

**Authors:** Minyue Qiu, Lixin Zhao, Junjie Zhang, Yalan Wang, Minchi Liu, Dong Hua, Xiaoyan Ding, Xiaoyang Zhou, Jie Zeng, Huacheng Yan, Jintao Li

**Affiliations:** 1Department of Biosafety, School of Basic Medicine, Army Medical University, Chongqing 400038, China; 18810467206@163.com (M.Q.); lxzhao98@163.com (L.Z.); zhangjj59@mail2.sysu.edu.cn (J.Z.); 2019010636@ybu.edu.cn (Y.W.); 15520085274@163.com (M.L.); 13919454081@163.com (D.H.); dxywork2012@sina.cn (X.D.); 15123188321@126.com (X.Z.); zengjie2019@yeah.net (J.Z.); 2Institute of Immunology, People’s Liberate Army, Army Medical University, Chongqing 400038, China; 3Center for Disease Control and Prevention of Southern Military Theatre, Guangzhou 510507, China; watshing@126.com

**Keywords:** dengue virus, intranasal route, neonate mice, A6 mice, DENV-2

## Abstract

Dengue virus, the causative agent of dengue fever, life-threatening hemorrhagic fever, and shock syndrome, is mainly transmitted to humans through mosquito vectors. It can also be transmitted through atypical routes, including needle stick injury, vertical transmission, blood transfusion, and organ transplantation. In addition, sporadic cases which have no clear infectious causes have raised the respiratory exposure concerns, and the risks remain unclear. Here, we analyze the respiratory infectivity of the dengue virus in BALB/c suckling and adult immunodeficient mice by the intranasal inoculation of dengue virus serotype 2. The infected mice presented with clinical symptoms, including excitement, emaciation, malaise, and death. Viremia was detected for 3 days post inoculation. Histopathological changes were observed in the brain, liver, and spleen. The virus showed evident brain tropism post inoculation and viral loads peaked at 7 days post inoculation. Furthermore, the virus was isolated from the infected mice; the sequence homology between the origin and isolates was 99.99%. Similar results were observed in adult IFN-α/β receptor-deficient mice. Overall, dengue virus can infect suckling mice and adult immune-deficient mice via the nasal route. This study broadens our perception of atypical dengue transmission routes and provides evidence of nasal transmission of dengue virus in the absence of mosquito vectors.

## 1. Introduction

The seasonal emergence of dengue virus (DENV) has been increasingly causing epidemics and has raised the concerns of the public health systems worldwide [1,2]. DENV, belonging to the family Flaviviridae, is mainly transmitted through mosquito bites. The pathogen causes self-limited flu-like dengue fever (DF) or life-threatening severe dengue, such as dengue hemorrhagic fever (DHF) and dengue shock syndrome (DSS) [3]. In the last 50 years, the morbidity associated with DENV infections has increased substantially, affecting four billion people across 128 countries [4]. Notably, besides the primary transmission route, atypical transmission routes, such as needle stick injury, vertical transmission and blood transfusion, or organ transplantation, have occasionally been documented since 1990 [5,6,7,8,9,10]. Sporadic cases that occurred in the absence of the vector are speculated to be caused by the respiratory transmission of DENV [11].

Recently, it was reported that other members of the family Flaviviridae, such as Zika and Japanese encephalitis virus (JEV), can infect guinea pigs or mice via the intranasal route [12,13]. Previous studies in West Nile virus, St. Louis encephalitis virus, and hepatitis C virus (HCV) have reported similar results [14,15,16]. However, the possibility of intranasal infection with DENV has not been confirmed or verified, which is intriguing because 35–38% of patients with DENV infection show respiratory symptoms, including sore throat, cough, and nasal congestion [17]. In addition, although rare, the virus has been isolated from the cerebrospinal fluid, saliva, urine, and upper respiratory tracts of patients with respiratory symptoms [18,19,20,21].

In this study, we identified the intranasal transmission of DENV under experimental conditions for the first time to facilitate a better understanding of non-vector-borne disease outbreaks in DENV endemic areas. To determine whether DENV can infect experimental mice through the intranasal route, BALB/c suckling mice, which are widely used in DENV research [22,23], and adult IFN-α/β receptor-deficient mice (A6 mice), which are susceptible to DENV [24], were exposed to dengue virus serotype 2 (DENV-2) through intranasal inoculation.

## 2. Methods

### 2.1. Cells and Viruses

Vero cells were cultured for 5–8 passages in DMEM medium (Gibco, New York, NY, USA) with 10% fetal bovine serum (FBS) and incubated in a humidified atmosphere containing 5% CO_2_ at 37 °C. The cells at 80–85% confluency were inoculated with DENV-2. DENV-2 was cryopreserved and provided by the Center for Disease Control and Prevention of the Southern Military Theatre. Three-day-old BALB/c suckling mice were inoculated with DENV-2 via the intracranial route, and the brain was harvested at 5 days post inoculation (dpi) and quickly ground in PBS on ice. The brain homogenate was collected and centrifuged at 10,000× *g* for 10 min. This procedure was repeated twice. The collected supernatants were stored as viral stocks at −80 °C. Titers were evaluated using the plaque assay with Vero cells.

### 2.2. Animal Infection

Healthy pregnant BALB/c mice weighing 30–35 g were purchased from the Animal Center of the Army Medical University (Third Military Medical University, Chongqing, China). They were provided with sterile water and chow ad libitum and acclimatized for 5–7 days before delivery. Three days after birth, suckling mice were intranasally inoculated with 3 μL of DENV-2 by pipetting to one nare. For the intracranially challenged group, mice were intracranially inoculated with 20 μL of DENV-2. Animals were monitored for clinical phenotypes, morbidity, and mortality. Clinical phenotypes were evaluated independently by three observers in a single-blinded manner. Moribund mice were euthanized with isoflurane and anatomized to harvest the heart, liver, spleen, lungs, kidneys, and brain. The tissue samples were fixed in 4% paraformaldehyde for 24 h at 4 °C, embedded in paraffin, and sectioned. To determine the viral kinetics, tissues and blood were harvested every other day from 1 dpi and ground in a homogenizer with precooled TRIzol (Invitrogen, Waltham, CA, USA). Then, tissues were quickly homogenized on ice for RNA isolation. Male A6 mice aged 6–8 weeks were intranasally challenged with 30 μL of DENV-2 by pipetting to one nare.

### 2.3. RNA Isolation and qRT-PCR

The total RNA was extracted from the heart, liver, spleen, lung, kidney, brain, and blood using TRIzol reagent (Tiangen Biotech, Beijing, China) according to the manufacturer’s recommendations. The PrimeScript^TM^ RT Reagent Kit (TaKaRa, Tokyo, Japan) was used to reverse-transcribe the total RNA to cDNA. Quantitative real-time reverse transcription-PCR (qRT-PCR) was performed using TB Green Premix Ex Taq II (TaKaRa, Tokyo, Japan) with the LightCycler^®^ 96 system (Roche, Basel, Switzerland). Serial dilutions of the DENV-2 plasmid with known concentrations were used to establish the standard curve and evaluate the amplification efficiency of the system. RNA copies per mL or RNA copies per gram of each sample were calculated from the quantification cycle (Cq) values using quantitative PCR with the standard curve. The relative expression levels of IL-1β, IL-6, TNFα, CCL2, CCL3, CCL4, and CCL5 were determined using RT-PCR. The primers used in this study are listed in Appendix A.

### 2.4. Immunofluorescence Staining

Paraffin-embedded mouse brain sections were dewaxed using a graded series of xylol and hydrated with a graded series of ethanol. Then, sections were incubated with 3% hydrogen peroxide solution for 25 min to reduce endogenous peroxidase activity. After that, sections were boiled in 0.01 M citrate buffer (pH 6.0) for 10 min to repair the antigens and incubated with normal goat serum (Abcam, Cambridge, UK) to block non-specific staining for 2 h at 37 °C. Next, sections were incubated with anti-dengue virus 1 + 2 + 3 + 4 primary antibodies (ab26837, Abcam, Cambridge, UK) at 4 °C overnight, and were subsequently incubated with FITC-conjugated secondary antibodies (Servicebio, Wuhan, China) for 2 h. Finally, sections were stained with 4,6-diamidino-2-phenylindole (DAPI) (ab228549, Abcam, Cambridge, UK) for 4 min. Representative sections were scanned using a Pannoramic DESK digital slice scanner (3D HISTECH, Budapest, Hungary) with the Pannoramic Scanner software.

### 2.5. Histopathology Staining

Paraffin-embedded tissue sections were dewaxed with xylol, hydrated with ethyl alcohol, and stained with haematoxylin for 3 min and eosin for 5 min. Images were photographed using a 40× objective with the Eclipse Ci-L microscope (Nikon, Tokyo, Japan) equipped with a DS-U3 camera (Nikon, Tokyo, Japan).

### 2.6. Transmission Electron Microscopy

Brain tissues were quickly trimmed into cuboids to a volume of ~1 × 1 × 2 mm^3^. The cuboids were fixed with 2.5% glutaraldehyde overnight at 4 °C and washed with PBS for 15 min three times. The cuboids were fixed again with 2% osmic acid for 2 h. Following graded acetone dehydration, tissues were infiltrated, embedded, and polymerized in resin. The resin mass was sliced to a thickness of 70 nm, and sections were stained with uranyl acetate and lead citrate and imaged using a JEM 1200EX (JEOL, Tokyo, Japan) transmission electron microscope.

### 2.7. Virus Isolation and Identification

Moribund suckling mice were sacrificed and anatomized to harvest their brains at 5 or 9 dpi (intracranially or intranasally inoculated mice, respectively). The brain was quickly ground with PBS and centrifuged to obtain the supernatant as isolates. Vero cells were transfected with 100-fold diluted isolates when the confluency reached 80%. After incubated for 2 h, Vero cells were maintained in DMEM medium (Gibco, Grand island, NY, USA) supplemented with 2% FBS (Gibco, Grand island, NY, USA) at 37 °C in an atmosphere with 5% CO_2_. Seven days later, the culture medium and cells were collected, and this constituted one passage. Additional passages were performed using the same method with 10-fold dilutions of the early generation viral stock. Viral RNA was detected using qRT-PCR, as described above, and viral titers were evaluated using the plaque assays.

### 2.8. DENV-2 Genome Sequencing and Analysis

The total viral RNA was extracted from DENV-2 or mock-infected mouse brains using the Purelink RNA minikit (Life Technology, Shanghai, China). Next, cDNA was generated using Moloney murine leukemia virus (Mo-MLV) reverse transcriptase (TaKaRa, Tokyo, Japan) with a reverse primer, and 15 primer pairs were used to generate overlapping amplicons spanning the entire genome accordingly. Sequencing was performed using the ABI 3730xl sequencer (ABI, Carlsbad, CA, USA). The genome sequence of DENV-2 was assembled by mapping the reads to the reference genome (Accession No. KM204118.1) of the DENV-2 New Guinea C strain using the DNA STAR v7.1 software (DNASTAR, Madison, WI, USA).

### 2.9. Plaque Assay

DENV-2 isolates were serially diluted with DMEM medium (Gibco, Grand island, NY, USA) from 10^−2^ to 10^−6^ times and added into six-well plates with Vero cell monolayers of 80–85% constancy. The plates were incubated at 37 °C for 2 h and gently shook every half hour. After removing the supernatant, 2 mL of semisolid maintenance medium (1.2% methylcellulose and 2% FBS in DMEM medium) was added to each well and incubated for 5–7 days at 37 °C. Subsequently, the semi-solid medium was removed carefully and 1% crystal violet stain (Beyotime, Shanghai, China) was added; the plates were incubated for 20–30 min. The plaques were counted after the plates were rinsed and air dried.

### 2.10. Ethics Statement

All animal and cell experiments were performed under biosafety level 2 conditions and conformed to the Chinese Regulations of Laboratory Animals. All experimental procedures were approved by the Ethics Committee for Animal Experimentation of the Army Medical University (AMUWEC20201490) and followed the guidelines of the National Institutes of Health Guide for the Care and Use of Laboratory Animals.

### 2.11. Data Analysis

All data were analyzed using GraphPad Prism v6.01 software (GraphPad Software, Inc., La Jolla, CA, USA). Data are shown as mean ± SD. The log-rank test was used for survival analysis. The two-way analysis of variance (ANOVA) followed by Tukey’s test was used to evaluate the significance between passages and viral origins. In all tests, values with *p* < 0.05 were considered significant.

## 3. Results

### 3.1. Suckling Mice Can Be Efficiently Infected by DENV-2 via the Intranasal Route

In order to confirm the possibility of intranasal transmission of DENV, suckling mice, which are widely used for virus fixation, isolation, virulence detection, and virulence recovery in DENV-related research, were used in this study (Figure 1A). As an intracranial challenge results in infection in suckling mice, we compared infectivity between intranasal and intracranial DENV inoculation in suckling mice. Three-day-old suckling mice were intranasally inoculated with 2.4 × 10^4^ plaque forming unit (PFU) of DENV-2, and the weight, clinical signs, and vital status of each animal were monitored thereafter. We noticed that in the intranasally challenged group, the infection symptoms usually started at 7–9 dpi, which was several days later than that in the intracranially challenged group. Moreover, all challenged mice showed the same clinical symptoms, including emaciation and malaise in the later stages (Figure 1B). Unexpectedly, some mice in the intranasally challenged group appeared to be excited by presenting with mania and sensitivity to moving objects before the fatal illness (Appendix A). Infections in both groups resulted in 100% mortality; however, the mean survival time of the intranasal inoculation group was prolonged from 6 to 9 days than the intracranial inoculation group (Figure 1C). Furthermore, we evaluated the dose-dependent survival rates for suckling mice in the intranasally challenged group; 100%, 62.5%, and 33.3% of the mice inoculated with 2.4 × 10^4^, 2.4 × 10^3^, and 2.4 × 10^2^ PFU of DENV-2 died, respectively (Figure 1D). Therefore, we selected a titer of 2.4 × 10^4^ PFU for subsequent experiments. In addition, the process of infection was consistent with the resulting weight changes in the mice. In the beginning, the weight change rates of mice in the challenge groups were substantially similar to those in the control group. However, once clinical signs were observed, the weight change rates significantly decreased in mice in the challenged groups (Appendix A).

### 3.2. Brain Is the Main Target of DENV-2 after Intranasal Inoculation

The blood, heart, liver, spleen, lung, kidney, and brain tissues of intranasally inoculated suckling mice were harvested and collected at 1, 3, 5, 7, 9, and 11 dpi to detect the viral loads. Owing to the shorter mean survival time, the blood and tissues of mice in the intracranially challenged group were collected at 1, 3, 5, and 6 dpi. In the intranasally challenged group, viremia was detected in all mice at 5 dpi by qRT-PCR, with peak loads of 3.55 × 10^4^ copies/mL (Figure 2A). Additionally, DENV-2 RNA was detected in multiple organs at 5 dpi using qRT-PCR, with peak loads ranging from 3 × 10^3^ to 7.59 × 10^3^ copies/g (Appendix A). In the brain tissues, DENV-2 RNA was detected at 3 dpi, and the virus replicated linearly, reaching a peak with an average viral load of 6.31 × 10^7^ copies/g at 7 dpi (Figure 2B), which was approximately twice as high as that in the other organs. Viremia and higher viral loads were observed 2 days earlier in the intracranially challenged mice than in the intranasally challenged mice, (Appendix A). Although DENV-2 replicated more rapidly in the brain tissues of intracranially challenged mice, resulting in an early peak in viral load (Figure 2B), the difference in the final RNA copy number between the two groups was not significant (Appendix A). In addition, immunofluorescence staining showed clear fluorescence signals in the brains of intranasally inoculated suckling mice at 9 dpi, when the clinical phenotype related to DENV-2 infection was the most obvious (Figure 2C). The results of electron microscopy analysis confirmed the presence of DENV-2 particles in the brain of suckling mice at 9 dpi (Figure 2D).

### 3.3. Histopathological Changes in Multiple Organs of Mice after Intranasal Inoculation

Next, we characterized the pathological changes that occurred in the challenged suckling mice. Despite the lower viral loads in the organs besides the brain, intranasal infection with DENV-2 resulted in substantial pathological changes in the liver and spleen of the moribund mice. The histopathological phenotypes were comparable between the intranasally and intracranially challenged mice with moderate inflammatory cell infiltration and the presence of a few multinucleated giant cells in the liver, as well as numerous apoptotic bodies and inflammatory cell infiltration in the spleen (Figure 3A). No obvious histopathological changes were observed in other organs (Appendix A). Histopathological changes in the brain tissue were observed in the hippocampus and cerebral cortex in both intracranially and intranasally challenged mice. In the intracranially challenged mice, a substantial number of necrotic pyramidal cells with pyknotic, broken, or dissolved nuclei, eosinophilic cytoplasm, and mild gliocyte proliferation were identified in all areas of the hippocampus (Figure 3B, Appendix A). Moderate pyramidal cell necrosis was observed in the CA2, CA3, and DG areas of the hippocampus in the intranasally challenged mice (Figure 3B, Appendix A). The predominant histopathological changes in the cerebral cortex were neuronal necrosis and vacuolization, which were mild in the intracranially challenged mice, but were profound in the intranasally challenged mice (Figure 3B).

### 3.4. DENV-2 Can Be Isolated from Intranasally Inoculated Mice and Efficiently Replicate in Vero Cells

After intracranial or intranasal inoculation, moribund mice were sacrificed and anatomized to harvest the brain tissue at 5 or 9 dpi, respectively (Figure 4A). DENV-2 in the supernatant of ground brain tissue from intracranially and intranasally challenged groups showed similar growth and plaque morphology to the viral stock (Figure 4B). Then, virus isolates of intracranially and intranasally challenged groups were serially passaged on Vero cells for three times. Cytopathic effects, viral RNA copy numbers, and infectious virion numbers were evaluated at each passage. The results showed that DENV-2 isolates from the two challenged groups could both induce cytopathic effects (CPE) in Vero cells within 5 days after inoculation (Appendix A) and effectively replicated resulting in high viral RNA loads (Figure 4C). Interestingly, as shown in Figure 4D, unlike the host adaptation process conducted by DENV-2 isolates from intracranially challenged mice, DENV-2 isolated from intranasally challenged ones could assemble and release greater numbers of mature virions in Vero cells since the first generation. Furthermore, the sequence homology between the viral stock and DENV-2 isolates was 99.99%, and the DENV-2 isolate showed 99.68% sequence homology with the DENV-2 New Guinea C strain (GenBank: KM204118.1) (Appendix A).

### 3.5. Adult A6 Mice Can Be Efficiently Infected by DENV-2 via the Intranasal Route

To validate the intranasal infectivity of DENV-2 in adult mice, groups of 4–6 weeks old male A6 mice were challenged with 2.4 × 10^5^ PFU of DENV-2. The weight, clinical signs, and vital status of each animal were recorded. The results showed that A6 mice could be infected with DENV-2 by intranasal inoculation. The clinical symptoms of intranasally infected A6 mice included emaciation, malaise, hind leg paralysis (Figure 5A), and death. At 7 dpi, 100% of the mice exhibited weight loss (Appendix A), and 50% of the mice were dead at 11–12 dpi (Figure 5B), with weight change rates about −30%. Interestingly, 50% of intranasally challenged mice showed recovery from neurological symptoms at 9–11 dpi, with regaining body weight (Appendix A). Moreover, after inoculation with higher viral titers, dose-dependent survival rates were observed (Appendix A). The study of moribund mice showed that intranasal infection causes mild and transient viremia in mice at 3–5 dpi (Figure 5C), probably due to limited migration of DENV-2 from the nasal passages to blood in adult mice. Robust viral replication and production of pro-inflammatory factors in the mouse brain (Figure 5D,E) indicated that the brain rather than the respiratory system may serve as the preferred target of DENV-2 infection following intranasal infection.

## 4. Discussion

The re-emergence of DENV is a major public health concern in tropical and subtropical regions. In the 1950s, *Aedes* mosquitoes were shown to play an important role in the transmission of DENV [25]. Subsequently, several atypical transmission routes for DENV have been characterized in humans, such as needlestick injury, vertical transmission and blood transfusion, or organs transplantation. However, rare cases [11,18] of suspected respiratory exposure to DENV were not stressed due to the lack of experimental confirmation and preconceived criticism regarding the failure of mosquito prevention programs [11,26]. In this study, we found that intranasal administration of DENV-2 under experimental conditions can infect both wild-type suckling mice and adult immunodeficient mice, with dose-dependent infection outcomes. As DENV infects and replicates in the human primary lung epithelium and lung cancer cell lines [27] and has been isolated from the upper respiratory tract of patients [18,19], our findings further strengthen the evidence for DENV infectivity in the upper respiratory mucosa in vivo.

Here, after intranasal inoculation, challenged mice showed systemic infection and neurological symptoms, such as excitement/hind leg paralysis, emaciation, malaise, and death. Viremia was detected in all mice and substantial histopathological changes were observed in multiple tissues. However, the results showed that the brain was the main target of DENV-2 infection after intranasal inoculation, rather than the lung, and possibly follows a different infection mechanism. As a part of the immune system, the nasal mucosa can contain inhaled antigens [28]. DENV enters and replicates within macrophages surrounding the nasal epithelium [29]. Subsequently, DENV-carrying macrophages may enter the blood and cause systemic infections. The copies of DENV-2 RNA were found in the brain earlier and in higher concentrations than in other organs or blood, highlighting the involvement of the respiratory tract in early DENV-2 infection. After invading into the upper respiratory tract mucosa, DENV-2 may enter the brain directly through the olfactory nerve, leading to massive viral replication and various neurological symptoms. Subsequently, the recruited leukocytes may increase the severity of mouse encephalitis owing to the Trojan horse effect [30,31]. Unexpectedly, lower numbers of PFUs were observed in the replications of isolates from intracranially inoculated mouse brains than intranasal ones (Figure 4D), whereas the virus sequences were not different. Host–virus interaction might affect viral proteins post translation modifications in different organs even in the same host. Viruses are obligate intracellular infectious agents that exploit host machinery to modify their viral proteins for survival. One of the key modifications is protein glycosylation [32]. As the secreted viruses are restricted in semisolid medium, the plaque forming must rely on the virus mediated cell-cell membrane fusion, which could be affected by the glycosylation modifications of virus [32]. Therefore, we hypothesized that the glycosylation modifications level of intracranial origin DENV-2 were blocked after replication in Vero cells and caused limited cell_-cell membrane fusion in Vero cells. Later, the glycosylation modifications of DENV-2 were increased with serial passaging, and the PFU increased accordingly, presenting a host adaption process.

We noted that the morbidity and mortality of murine models varied after challenge with different titers of DENV-2, whereas the mice challenged with the same titer exhibited varying degrees of infection. This result suggests that the infectivity of DENV is influenced by virus titer and host immune responses. In addition to respiratory exposure, sexual transmission of DENV has been suspected [33]. There have been isolated case reports describing the sexual transmission of DENV in people from non-pandemic areas [34]. This finding indicates that the natural mucous membrane-dependent transmission route is feasible but rarely recognized. There are several potential explanations for this finding. First, viable virus titers in body fluids other than the blood may exist in a few cases, and these require closer examination [18]. Second, the ubiquity of mosquito vectors in epidemic areas [35] makes it difficult to exclude the possibility of a primary transmission route in potential cases. In addition, the 3–14-day incubation period of DENV and the poorly understood respiratory transmission mechanism makes the epidemiological investigation more difficult [36]. Importantly, people may have ignored the potential of alternate transmission routes for DENV, such as intranasal, aerosol, and sexual transmission routes, which may explain why the majority of atypical transmission cases occur in hospitals, laboratories, and non-endemic areas where mosquito control strategies are well implemented [37,38,39].

This study had several limitations. First, the mechanism of how DENV migrates from the upper respiratory tract to the brain remains unclear. The blood-brain barrier (BBB) is a complex structure that constitutes a barrier between the blood and the central nervous system [40]. BBB destruction caused by DENV infection may lead to encephalopathy or encephalitis, which is associated with morbidity in 5–7% of patients and mortality in ~50% of patients [41]. DENV-associated encephalopathy or encephalitis will be better understood if the mechanisms causing the upper respiratory tract-to-brain infection are further investigated. Second, to fully understand the potential for respiratory transmission of DENV, other nasal routes, such as inhalation of DENV droplets or aerosols, should be analyzed in animal models. Finally, the infectious titers of DENV used in this study are suitable for animal models alone; it remains unknown whether DENV can be transmitted in human populations through the nasal route. As described above, it is difficult to accurately identify the cause of DENV infection in individuals living in pandemic areas. With progress in DENV prevention and control programs [42,43], there may be an increase in non-vector dengue cases in the future.

In conclusion, this study showed that DENV-2 can cause acute infection in both suckling mice and adult A6 mice via intranasal inoculation, resulting in neurological symptoms. Upper respiratory tract cells may be the first site for virus invasion and replication, and the mechanisms underlying the upper respiratory tract-to-brain infection require further investigation. These findings indicate that at appropriate titers DENV-2 can infiltrate the respiratory mucosa of mice and cause infection, suggesting that the potential for respiratory transmission of DENV may be greater than that previously assumed. For cases of non-vector transmitted DENV infection, the likelihood of inhalation or contact with contagious droplets or aerosols should be considered, especially among healthcare workers who may be exposed to concentrated DENV contaminants.

## Figures and Tables

**Figure 1 viruses-14-01394-f001:**
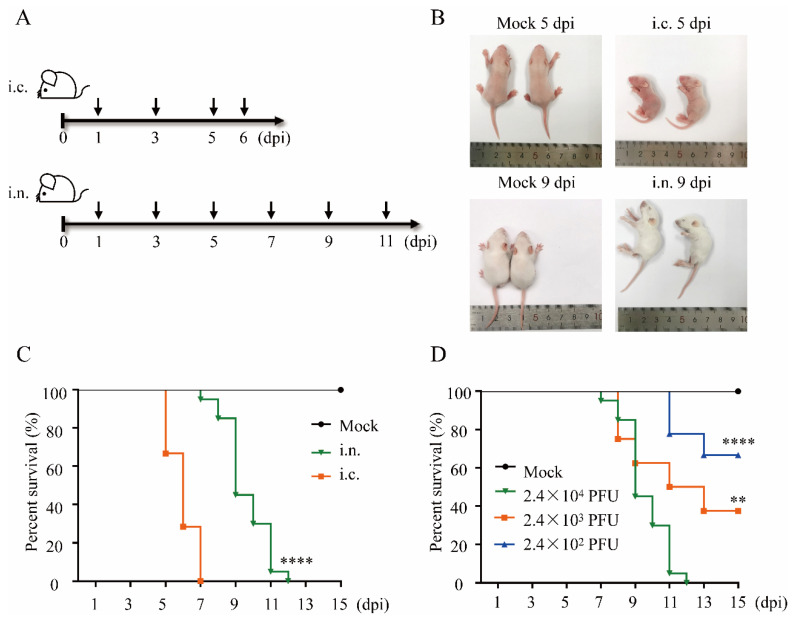
DENV-2 infection via the intranasal route is lethal in BALB/c suckling mice. (**A**) Experimental scheme. Three days after birth, the newborn BALB/c suckling mice were challenged intracranially or intranasally with DENV-2. The body weight, clinical phenotypes, and mortality rates were evaluated. The organs (heart, liver, spleen, lung, kidney, and brain) and blood of the challenged suckling mice were harvested (black arrow). (**B**) Representative images of intracranially or intranasally challenged BALB/c suckling mice. (**C**) Survival probability of DENV-2 challenged suckling mice infected via intracranial or intranasal routes. Survival conditions were monitored daily after challenge (titer = 2.4 × 10^4^ PFU; i.c., n = 21; i.n., n = 20; mock, n = 8). (**D**) Survival probability of intranasally infected suckling mice with different titers of DENV-2. Abbreviations: i.c., intracranial; i.n., intranasal; dpi, day post inoculation; PFU, plaque forming unit. Log rank test; ****, *p* < 0.0001; **, *p* < 0.01.

**Figure 2 viruses-14-01394-f002:**
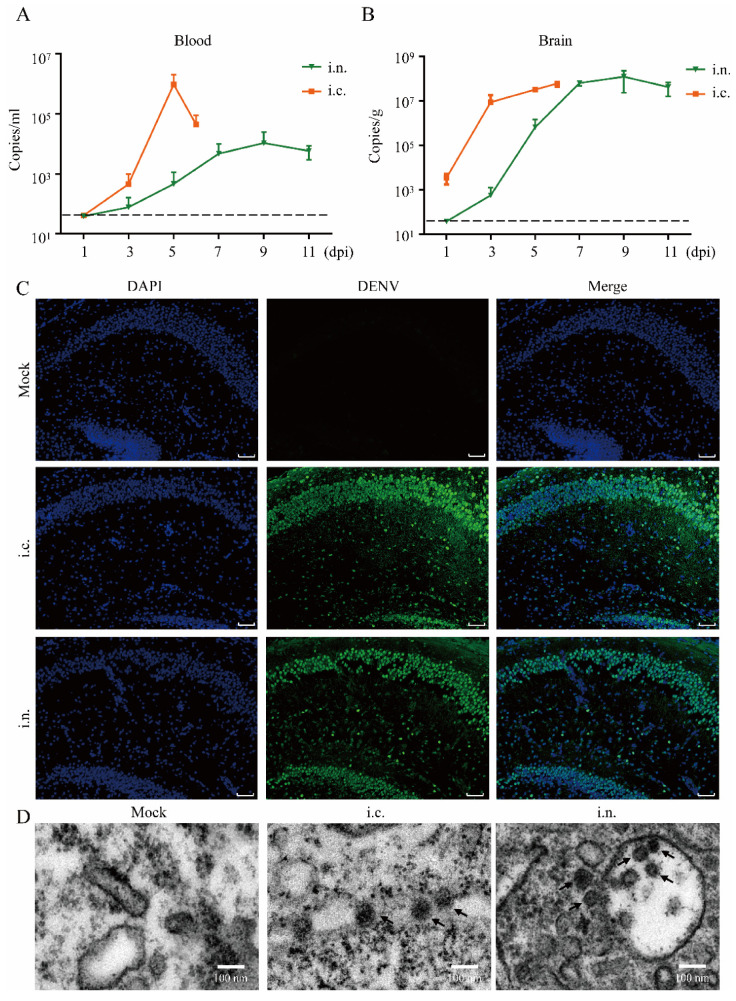
Infection status of DENV-2 challenged suckling mice. (**A**) Blood samples were harvested, and viremia was evaluated (each time point, n = 4 or n = 5). The dotted black line indicates the limits of detection. (**B**) Brain samples were harvested, and the virus titer was evaluated using RT-PCR (each time point, n = 4 or n = 5). (**C**) Immunostaining of the hippocampal CA1 and CA2 area from intracranially and intranasally challenged mice at the severe illness phase (5 and 9 dpi, respectively). Scale bar: 50 μm. (**D**) Representative transmission electron micrographs showing viral particles in DENV-2 challenged suckling mouse brains. Black arrows, viral particles. Abbreviations: i.c., intracranial; i.n., intranasal; dpi, day post inoculation.

**Figure 3 viruses-14-01394-f003:**
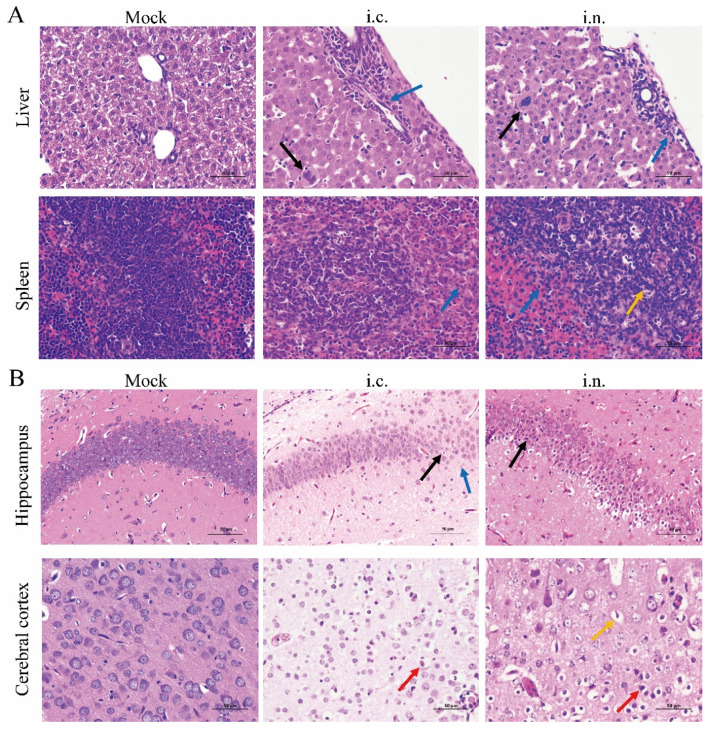
DENV-2 infection results in pathological changes. (**A**) Selected organs were harvested from intracranially or intranasally challenged mice sacrificed in a late phase of illness (5 and 9 dpi, respectively) and stained with hematoxylin and eosin (H & E). Blue arrows, granulocytes; black arrows, multinucleated giant cells; yellow arrows, apoptotic bodies. (**B**) Brains were harvested from intracranially or intranasally challenged mice sacrificed in the late phases of illness (5 and 9 dpi, respectively) and stained with H & E. Blue arrows, gliocyte proliferation; black arrows, necrotic pyramidal cells; red arrows, necrotic neurons; yellow arrows, cytoplasmic vacuoles. Scale bar: 50 μm. Abbreviations: i.c., intracranial; i.n., intranasal.

**Figure 4 viruses-14-01394-f004:**
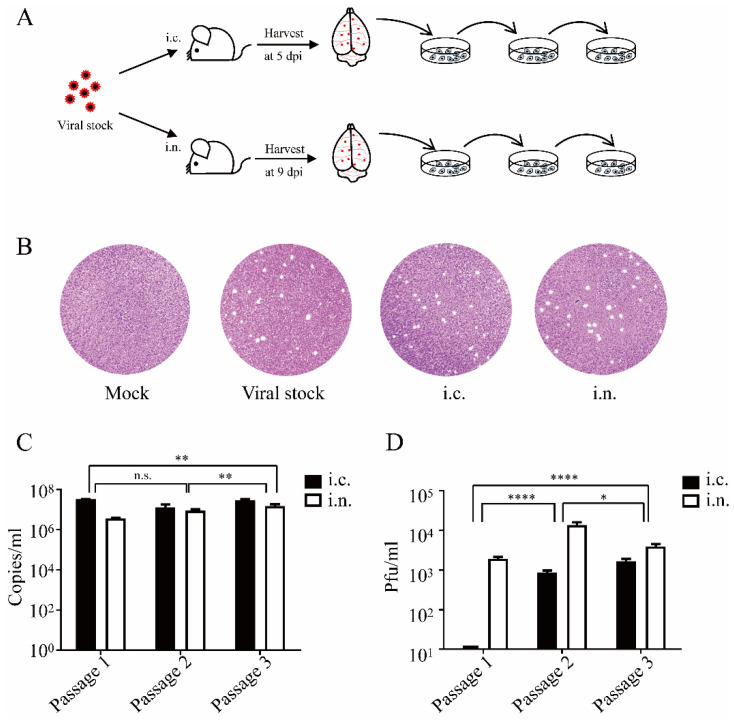
Isolation of DENV-2 from Vero cells. (**A**) Experimental scheme. Three days old BALB/c suckling mice were intracranially or intranasally challenged with DENV-2. The brain tissues were harvested at 5 or 9 dpi. Isolates were passage cultured for three generations. (**B**) Plaque morphology of DENV-2 viral origin and isolates from intracranially or intranasally infected mouse. (**C**) DENV-2 was isolated from Vero cells and cultured for three passages; viral loads were evaluated by RT-qPCR. (**D**) The viral titer of each passage was evaluated in Vero cells. Abbreviations: i.c., intracranial; i.n., intranasal; PFU, plaque forming unit. Histograms: mean ± S.D. Two-way ANOVA test; ****, *p* < 0.0001; **, *p* < 0.01; *, *p* < 0.05; ns, not significant.

**Figure 5 viruses-14-01394-f005:**
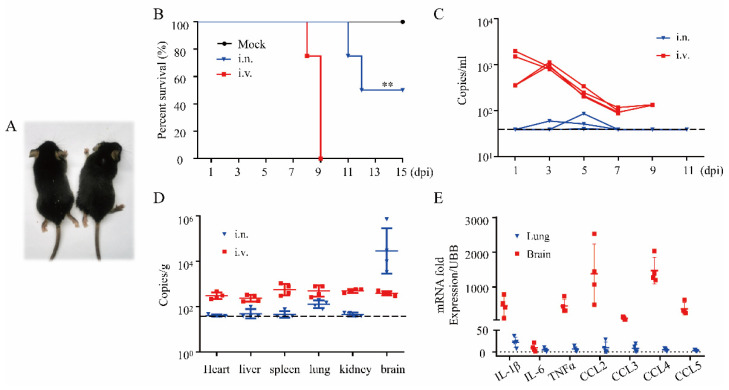
DENV-2 infection via the intranasal route is lethal in A6 mice. (**A**) Representative images of intranasally challenged A6 mice with neurological symptoms. (**B**) Survival probability of DENV-2 challenged A6 mice administered the virus via the intravenous or intranasal routes. Survival conditions were monitored daily after challenge (titer = 2.4 × 10^5^ PFU; i.v., n = 4; i.n., n = 4; mock, n = 4). **, *p* < 0.01. (**C**) Viremia in A6 mice after DENV-2 inoculation. (i.v., n = 4; i.n., n = 4). Dotted lines indicate the limits of detection. (**D**) Internal organ samples were harvested, and the virus titer was evaluated (i.v., n = 4; i.n., n = 4). (**E**) Expression of pro-inflammatory factors in the brains and lungs of moribund A6 mice. Abbreviations: i.v., intravenous; i.n., intranasal; dpi, day post inoculation.

## Data Availability

This study did not report any data.

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
