# Peer review of "Effective Infection with Dengue Virus in Experimental Neonate and Adult Mice through the Intranasal Route"

_viruses, 2022, doi:10.3390/v14071394_

Round 1
Reviewer 1 Report
The authors describe experiments to determine if dengue virus can be transmitted via the intranasal route thereby accounting for cases with no known vector exposure. The experimental design is appropriate and the authors demonstrate that both suckling mice and immune-deficient mice do become infected following intranasal administration of DENV-2 when administered at high PFU concentrations. The use of juvenile and immunodeficient mice is justified given that adult immune competent mice are refractory to DENV-2 infection. While intriguing, the implications of these results on the epidemiology of DENV-2 is questionable given that DENV-2 is found in saliva only rarely and likely at low titers. Still, the results to contribute to the scientific knowledge and support previous studies with other flaviviruses.
Other items to be addressed include: please describe the manner in which intranasal administration was accomplished. For example: volume of inoculum delivered via pipet to one or both nares? No need to place the in front of dengue virus. Common abbreviation for dengue virus serotype 2 is DENV-2. Figure 1 abbreviations need to be defined. Figure 3: arrows are not described for panel B. Particularly the red arrows. Figure 4, the difference between B and C is unclear. The abbreviations also need to be defined. Figure 5B = survival probability?
Author Response
We thank you for reviewing our manuscript and providing valuable comments and suggestions. We have revised the manuscript according to your comments and suggestions. Our responses to the comments are provided below.
Point 1: Please describe the manner in which intranasal administration was accomplished. For example: volume of inoculum delivered via pipet to one or both nares?
Response 1: We have added the description of intranasal administration manners in the manuscript (2. Methods; 2.2 Animal Infection).
Point 2: No need to place the in front of dengue virus. Common abbreviation for dengue virus serotype 2 is DENV-2.
Response 2: We have made the necessary revisions throughout the manuscript, including the abstract.
Point 3: Figure 1 abbreviations need to be defined.
Response 3: We have defined the abbreviations in the revised manuscript (Figure 1).
Point 4: Arrows are not described for panel B. Particularly the red arrows.
Response 4: We have added a description of the arrows in the revised manuscript (Figure 3).
Point 5: Figure 4, the difference between B and C is unclear. The abbreviations also need to be defined.
Response 5: The Y-axis title was incorrect; We have corrected the Y-axis title in Figure 4C (now is figure 4D) and defined the abbreviations in the revised manuscript (Figure 4).
Point 6: Figure 5B = survival probability?
Response 6: We thank you for your suggestion. We have changed the diction from “Survival proportion” to “Survival probability” throughout the manuscript, including the abstract.
Reviewer 2 Report
The authors propose an alternative, respiratory route of transmission for DENV2. There work establishes the capacity for intranasal infection in suckling mouse and immune deficient mouse model systems. The biological relevance for respiratory transmission in humans remains to be established, but this study lays groundwork for further investigations of transmission in humans where vector transmission can be excluded. Sufficient data is provided to confirm virus distribution to tissue, including brain, upon intranasal administration of virus in a dose-dependent manner. However, several experiments are not sufficiently described in the text and figure legends to allow interpretation, and images in figures 2 and 3 are of insufficient magnification and resolution to fully support the conclusions described.
What is “partial intranasal challenge”? There is no mention of partial challenge in table S1. Does this refer to animals infected at lower doses?
The symptoms don’t appear in “early stages”. In table S1, symptoms appear to start at 7 and 8 dpi of a 12-day infection. Is 7-8 days considered an early stage?
Please express genome copies/ mL in scientific notation, ie. Copies/mL 101, 102, 103 etc, not as “log(copies)/ml.
What are the structures in hippocampus in figure 2C? The region at the top in the images has high cell density and complete infection. What specific antigens are labeled by the Abcam antibodies. Do you have a section with control antibody?
In figure 2D, explain in the legend that red arrowheads indicate virus particles. A black arrow with a tail would be better at seeing where the point is and the point should not impinge on the virus particle. A higher magnification with a 100nM scale bar would be more convincing. As it is, the particles do not have sufficient resolution to be confirmatory, only their juxtaposition in a vesicle in the middle panel supports identification as virus particles. In the third panel, there is a second particle that appears in a vessel or membrane that is not marked with an arrowhead. Is it the intent that that image is not a virus particle?
Interpretation of Figure 3 is also hampered by low magnification and low resolution of the images. Characterization of intracellular features requires higher magnification. The legend does not explain what feature is indicated by the yellow arrows. Why don’t the hippocampus sections correspond to those used in the IFAs in figure 2 where the mantle is highly enriched with DENV antigen. Does this region have higher infiltration of inflammatory cells or other indicators of pathology? The text refers to pyramidal cell necrosis in specific areas of the hippocampus, but these images are not shown.
In figure 4B, it is unclear what significance passage in Vero cells has. If the virus did not change during passage in mice, why would it change upon serial passage in Veros? Likewise, the significance and results in figure 4C are unclear. From the text and the legend, it is not clear what the difference is between virus passage in figures 4B and 4C. The text indicates that figure 4C somehow indicates “numbers of mature virions”. The figure legend indicates “viral titre”, which is usually measured in PFUs, but 4C is in genome copy numbers just like 4B. Was it supposed to be plaque numbers in 4C? Were the passages in 4C plaqued on Veros or passaged in Veros and measured by RT-PCR or passaged in mice and plaqued on Veros to assess “mature” particles? Does the phrase “Mature Particle” mean infectious particle as opposed to genome copy? Is figure 4C meant to show that initial virus preps from mice after intracranial administration infected Veros poorly and that their “maturation” or ability to infect Veros was increased by passage in Veros and that the intranasally infected animals yielded more infectious particles the initial isolation?
On which days post infection, exactly, were the ground brain preps made? The text indicates “with the highest load”. Figure 2B indicates that this might be day 6 or 7 for both inoculation methods, but would be late for IC in section and early for IN infection. The timing of harvest post infection may affect the yield of infectious particles, depending on host control of particle formation, ie innate and adaptive targeting on virus particle infectivity.
Please clarify these points, including precise descriptions of the parameters of the experiments in Figure 4.
Author Response
We thank you for reviewing our manuscript and providing valuable comments and suggestions. We have revised the manuscript according to your comments and suggestions. Our responses to the comments are provided below.
Point 1. What is “partial intranasal challenge”? There is no mention of partial challenge in table S1. Does this refer to animals infected at lower doses?
Response 1: We have rewritten the relevant sentence as following to improve the clarity and readability: “some mice in the intranasally challenged group appeared to be excited by presenting with mania and sensitivity to moving objects before the fatal illness.”
Point 2. The symptoms don’t appear in “early stages”. In table S1, symptoms appear to start at 7 and 8 dpi of a 12-day infection. Is 7-8 days considered an early stage?
Response 2: There were individual differences in the onset and survival times after DENV infection. “Early stages” refers to the mild stage in each subject. We have rewritten the relevant sentence as response 1 to improve the clarity and readability.
Point 3. Please express genome copies/ mL in scientific notation, ie. Copies/mL 101, 102, 103 etc, not as “log(copies)/ml.
Response 3: We have made the necessary revisions throughout the manuscript, including the abstract.
Point 4. What are the structures in hippocampus in figure 2C? The region at the top in the images has high cell density and complete infection.
Response 4: The structures showed in Figure 2C is the hippocampal CA1 and CA2 area. We have added a descriptions of the structures in the revised manuscript (Figure 2).
Point 5. What specific antigens are labeled by the Abcam antibodies. Do you have a section with control antibody?
Response 5: As mentioned in the product datasheet of ab26837, the immunogens of the antibody are viral particles from C6/36 amplified dengue type 2 virus and the antibody is rabbit polyclonal to Dengue Virus 1+2+3+4. We have verified the specificity of the antibody on uninfected animals (Mock). The results are demonstrated in Figure 2C.
Point 6. In figure 2D, explain in the legend that red arrowheads indicate virus particles.
Response 6: We have added a description of the arrows in the revised manuscript (Figure 2D).
Point 7. A black arrow with a tail would be better at seeing where the point is.
Response 7: We thank you for your suggestion. We have changed the arrows to black arrows with tails accordingly in figure 2D.
Point 8. The point should not impinge on the virus particle. (figure 2D)
Response 8: We have ensured that the arrows are not impinging the virus particles in the revised manuscript.
Point 9. A higher magnification with a 100 nM scale bar would be more convincing. As it is, the particles do not have sufficient resolution to be confirmatory, only their juxtaposition in a vesicle in the middle panel supports identification as virus particles. (figure 2D)
Response 9: The low resolution was caused by image compression during the export. We have ensured that all the figures have been re-exported with high definition. Images with a 100 nM scale bar have been replaced in the revised manuscript. We have rechecked the images of the samples, and there was no juxtaposition of virus particles in a vesicle in the middle panel in this sample. However, the images were evaluated independently by three professional experts in a single-blinded manner. They all recognized the virus particles marked in the images. According to the suggestion of the experts, we have replaced the images with images which have better views from the same batch in the revised manuscript.
Point 10. In the third panel, there is a second particle that appears in a vessel or membrane that is not marked with an arrowhead. Is it the intent that that image is not a virus particle? (figure 2D)
Response 10: We have rechecked the image and we think the second particle is a virus particle. To show the result better, we have replaced the images with other images from the same batch which have a better view to show the virus particles.
Point 11. Interpretation of Figure 3 is also hampered by low magnification and low resolution of the images. Characterization of intracellular features requires higher magnification.
Response 11: The magnification of the images in figure 3 is 400 ×. The low resolution was caused by image compression during the export. We have ensured that all the figures were re-exported at a high definition and replaced in the manuscript.
Point 12. The legend does not explain what feature is indicated by the yellow arrows. (Figure 3)
Response 12: We have revised the description of the arrows in the revised manuscript (Figure 3).
Point 13. Why don’t the hippocampus sections correspond to those used in the IFAs in figure 2 where the mantle is highly enriched with DENV antigen. Does this region have higher infiltration of inflammatory cells or other indicators of pathology? (Figure 3)
Response 13: The viral fluorescence signal was highly expressed in all regions of the hippocampus. Here we chose the areas that have the most obvious pathological changes and collected the images for each sample. To fully display the histopathological changes of hippocampus, we have realigned the images of different area of hippocampus. Now the images in figure 3 are hippocampal CA1 and CA2, while the images of CA3 and DG area are shown in supplementary figure 4 in the revised manuscript.
Point 14. The text refers to pyramidal cell necrosis in specific areas of the hippocampus, but these images are not shown.
Response 14: We have added the image of other areas of hippocampus in the revised manuscript (Supplementary Figure S4).
Point 15. In figure 4B, it is unclear what significance passage in Vero cells has. If the virus did not change during passage in mice, why would it change upon serial passage in Veros?
Response 15: The reason why we passaged the isolated virus in Vero cells is to prove that DENV-2 can not only infect suckling mice through the intranasal route, but can also be isolated from the infected mice brain tissues and passaged stably for several generations. We assumed that these results could reconfirm the intranasal infectivity of DENV-2 in the experimental animals. We have added an experimental scheme in figure 4 in the revised manuscript, to show this part clearer (Figure 4A).
Point 16. The significance and results in figure 4C are unclear. From the text and the legend, it is not clear what the difference is between virus passage in figures 4B and 4C. The text indicates that figure 4C somehow indicates “numbers of mature virions”. The figure legend indicates “viral titer”, which is usually measured in PFUs, but 4C is in genome copy numbers just like 4B. Was it supposed to be plaque numbers in 4C? Were the passages in 4C plaqued on Veros or passaged in Veros and measured by RT-PCR or passaged in mice and plaqued on Veros to assess “mature” particles? Does the phrase “Mature Particle” mean infectious particle as opposed to genome copy?
Response 16: It is the plaque numbers, as determined using the plaque assay, in original figure 4C (now is figure 4D). The Y-axis title was incorrect; it has been corrected in the revised manuscript.
Point 17. Is figure 4C meant to show that initial virus preps from mice after intracranial administration infected Veros poorly and that their “maturation” or ability to infect Veros was increased by passage in Veros and that the intranasally infected animals yielded more infectious particles the initial isolation?
Response 17: Yes, Figure 4C meant that. We have revised the description of Figure 4 in the revised manuscript (Results 3.4).
Point 18. On which days post infection, exactly, were the ground brain preps made? The text indicates “with the highest load”. Figure 2B indicates that this might be day 6 or 7 for both inoculation methods, but would be late for IC in section and early for IN infection. The timing of harvest post infection may affect the yield of infectious particles, depending on host control of particle formation, ie innate and adaptive targeting on virus particle infectivity.
Response 18: The ground brain preps were prepared at 5 dpi for intracranially inoculated mice and 9 dpi for intranasally inoculated mice. We have added the information in the revised manuscript (Results 3.4).
Point 19. Please clarify these points, including precise descriptions of the parameters of the experiments in Figure 4.
Response 19: We thank you for your suggestions. We have revised the manuscript according to the comments and suggestions. The description of figure 4 has been rewritten in the revised manuscript (Results 3.4).
Round 2
Reviewer 2 Report
Response to review is satisfactory, however, the authors should offer possible explanations for the reduced numbers of PFUs observed for the isolates from IC-inoculated mouse brains in the discussion.
Author Response
Point 1: the authors should offer possible explanations for the reduced numbers of PFUs observed for the isolates from IC-inoculated mouse brains in the discussion.
Response 1: Unexpectedly, lower numbers of PFUs were observed in the replication of isolates from intracranially inoculated mouse brains than intranasal ones (Fig. 4D), whereas the virus sequences were not different. Host–virus interaction might affect viral proteins post translation modifications in different organs even in the same host. Viruses are obligate intracellular infectious agents that exploit host machinery to modify their viral proteins for survival. One of the key modifications is protein glycosylation [32]. As the secreted viruses are restricted in semisolid medium, the plaque formation must rely on the virus-mediated cell–cell membrane fusion, which could be effected by the glycosylation modifications of virus [32]. Therefore, we hypothesized that the glycosylation modifications of intracranial origin DENV-2 were blocked after replication in Vero cells and caused limited cell–cell membrane fusion in Vero cells. Later, the glycosylation modifications of DENV-2 were increased with serial passaging, and the PFU increased accordingly, presenting a host adaption process.
More details please see the attachment.
